# Nutrition in Abrupt Sunlight Reduction Scenarios: Envisioning Feasible Balanced Diets on Resilient Foods

**DOI:** 10.3390/nu14030492

**Published:** 2022-01-23

**Authors:** Alix Pham, Juan B. García Martínez, Vojtech Brynych, Ratheka Stormbjorne, Joshua M. Pearce, David C. Denkenberger

**Affiliations:** 1Alliance to Feed the Earth in Disasters (ALLFED), Fairbanks, AK 99775, USA; juan@allfed.info (J.B.G.M.); vojtech@allfed.info (V.B.); ratheka.stormbjorne@gmail.com (R.S.); joshua.pearce@uwo.ca (J.M.P.); ddenkenberger@alaska.edu (D.C.D.); 2Department of Electrical & Computer Engineering, Western University, London, ON N6A 5B9, Canada; 3Department of Mechanical Engineering, University of Alaska Fairbanks, Fairbanks, AK 99775, USA; 4Alaska Center for Energy and Power, University of Alaska Fairbanks, Fairbanks, AK 99775, USA

**Keywords:** global catastrophic risk, existential risk, nutrition, food security, nuclear winter, resilient food

## Abstract

Abrupt sunlight reduction scenarios (ASRS) following catastrophic events, such as a nuclear war, a large volcanic eruption or an asteroid strike, could prompt global agricultural collapse. There are low-cost foods that could be made available in an ASRS: resilient foods. Nutritionally adequate combinations of these resilient foods are investigated for different stages of a scenario with an effective response, based on existing technology. While macro- and micronutrient requirements were overall met, some—potentially chronic—deficiencies were identified (e.g., vitamins D, E and K). Resilient sources of micronutrients for mitigating these and other potential deficiencies are presented. The results of this analysis suggest that no life-threatening micronutrient deficiencies or excesses would necessarily be present given preparation to deploy resilient foods and an effective response. Careful preparedness and planning—such as stock management and resilient food production ramp-up—is indispensable for an effective response that not only allows for fulfilling people’s energy requirements, but also prevents severe malnutrition.

## 1. Introduction

Supply chain disruptions caused by the COVID-19 pandemic have emphasized the fragility of food systems as of late, serving as a scaled-down example of the potential severity of a global food system shock [1,2,3,4]. There currently is a clear need for more work on preparing for the most extreme and abrupt food system shocks. The global food production system, largely based on agriculture, is dependent on consistent environmental conditions, such as sunlight, temperature and precipitation, which all can be severely affected by both natural and anthropogenic factors. Potentially very severe risks to global food production that have been described in the food security literature are numerous, including crop pathogens, herbicide-resistant weeds, extreme crop pests, super bacteria displacing beneficial bacteria, abrupt climate change, slow but extreme climate change, mass pollinator decline or a combination of these [5,6]. Anthropogenic climate change is already intensifying the risk of concurrent severe weather events [7], negatively impacting food systems [8] and making it increasingly likely that concurrent shocks would result in a multiple bread-basket failure (MBBF) [9,10]. The ensuing significant crop yield declines would cause food prices to spike, thus exacerbating food insecurity [11].

This would be a dramatic scenario, but unfortunately there exist even more severe food-related global catastrophic risks (GCRs) than the aforementioned. A common definition of a GCR is “an event that could damage human well-being on a global scale, even endangering or destroying modern civilization” [12]. The current agriculture-based global food system is known to function in “self-undermining, self-debilitating dynamics, disrupting yields and supply chains”, thus sustaining a significant vulnerability to global food catastrophe [13]. We classified a subset of these most extreme food catastrophes under the label of abrupt sunlight reduction scenarios (ASRS), in which a sudden catastrophic event causes an immense amount of aerosol material of sulfates or black carbon (soot) to be projected and entrapped in the stratosphere for several years. The ensuing abrupt reduction in sunlight irradiation, global temperatures, and precipitation levels could swiftly trigger near-total global agricultural collapse, which would most likely push billions of people into starvation [14], given a lack of proper planning and preparedness. At least three potential mechanisms for such a catastrophe have been identified in the literature: a volcanic winter caused by a large volcanic eruption, a direct impact of an exceptionally large asteroid or comet, and a nuclear winter triggered by a nuclear war in which numerous cities have been targeted [5,12].

In the absence of significant preparation prior to one of these extreme global catastrophes taking place, there would be little opportunity for adaptation. Given there currently are enough food stocks to feed the global population for only a few months [15], and these catastrophes would likely last several years at a minimum, humanity would have to react quickly to prevent mass starvation globally. Some researchers estimate that in a full-scale nuclear winter scenario, around 75% of the global population would starve to death [14] in the absence of a rapid and effective response; and millions would likely starve from even moderate nuclear autumn from a smaller nuclear exchange [16]. Increasing planning and preparedness capabilities for scaling up food production methods that are resilient to these catastrophes has been proposed as a cost-effective solution to these GCRs [17]. This study aims at assessing the nutrition landscape in such a scenario, in order to better understand where planning and preparedness efforts should be focused to avoid massive famines, forms of malnutrition and global public health issues. 

A portfolio of food solutions that are resilient to abrupt catastrophic disruptions in the food system would be a valuable resource for increasing the resilience of food systems and preparedness globally. We define resilient foods or resilient food solutions as those foods, food production methods or interventions that would allow for significant food availability in the face of a global catastrophic food shock. These solutions should be well-suited for contributing to an adequate food supply for the greatest number of people even in the worst scenarios. Thus, this study focuses on technically-viable solutions that are both scalable and open source whenever possible to facilitate rapid production ramping. A preliminary version of such a resilient food portfolio is proposed in Section 2.3, with a focus on abrupt sunlight reduction scenarios. 

Affordability should also be a key factor if the majority of the population is to have adequate access to food during a global catastrophe, just as it is today [18]. Low-income populations in developing countries would expectedly be most at risk of starvation [19]. Even today, about 25,000 people die daily from hunger and related diseases [20]. Estimating food prices with a reasonable degree of precision during an extreme food catastrophe is, however, a formidable task, and strongly dependent on the details of the scenario. Nonetheless, we hypothesize that foods that are currently low cost are more likely to be available for the most underprivileged in the considered scenarios. 

Another key consideration is for these food solutions to be able to contribute to a diet as nutritionally balanced as possible during the catastrophe period to prevent malnutrition, the topic of the current work. A previous study on the topic [21] proved how a balanced diet can be achieved using only resilient food sources that could be produced during an ASRS, while warning that access to such a diet may not be available to the poorest populations. In contrast, this analysis aims to improve upon these results by demonstrating how the proposed portfolio of low-cost resilient foods can also be nutritionally adequate for an ASRS, using an existing nuclear winter model [22] as an example. To do this, the nutritional needs were estimated for the global human population and then compared to the micronutrient profiles and ramp rates of potentially available traditional and alternative resilient foods. The availability was divided into three periods: (I) food stocks (vegetable and animal), (II) potatoes and traditional resilient foods ramp-up, and (III) greenhouses, crop relocation, and alternative resilient foods. Both optimistic and pessimistic diets combining resilient foods available simultaneously—according to the proposed chronology—were created and then compared to needs of proteins, fats, carbohydrates, minerals, and vitamins. The results are presented and discussed in the context of improving humanity’s resilience to major food shocks.

## 2. Methods

### 2.1. Human Nutritional Needs

For simplicity, the energy needs to be covered are based on the minimum recommended requirements of an average adult as proposed by the World Health Organization (WHO) for food management during emergencies [23]: 2100 kcal of energy intake, for a body weight of 62 kg [24], as carried out in previous food-related GCR assessments [25,26,27,28].

The intake of each nutrient was classified into three different categories based on the nutrition literature: adequate intake, moderate- and severe-risk-associated intakes (Figure 1). Complete referencing is available in the Appendix A.

#### 2.1.1. Adequate Intakes (AI)

Lower adequate intake (AI) values were usually defined as the most conservative value between the dietary reference intake (DRIs) as defined by the United States Department of Agriculture (USDA) [29,30], the dietary reference values (DRVs) as defined by the European Food Safety Authority (EFSA) [31,32] and the population nutrient intake goals (PNIGs) from the WHO recommendations [34,35]. Upper AI values were derived in a similar manner from tolerable upper limits (TULs) defined by the USDA [30], the EFSA [33] and the WHO [34,35].

#### 2.1.2. Moderate- and Severe-Risk-Associated Intakes (MRAI and SRAI)

Depending on the risk associated with out-of-range nutrient intakes, they were further classified into moderate- and severe-risk-associated intakes (MRAI and SRAI). A similar rationale as for AIs was used to define the limit between MRAI and SRAI (for both upper and lower limits), using the second most conservative value if available. When more relevant or up-to-date data were found in the literature, such as in WHO emergency management handbooks [23,36], those values were used instead (see Appendix A ‘Human Nutritional Needs’ in the Appendix A). Alternatively, those values were used as the upper limit for SRAIs, if they were associated with extremely severe health risks—e.g., poisoning, long-term severe damage, or death.

### 2.2. Catastrophic Scenario Characterization

Post-disaster scenario analysis was based on the Coupe et al. [22,37] in-depth nuclear winter climate model. In their study, 150 Tg of soot are projected above the clouds, where rain cannot wash it out [22,38]. The subsequent climatic consequences, such as large drops in temperature and precipitation levels, were used as a basis to understand how the food system might be altered in this scenario, rather than as direct mathematical inputs. For example, it was used to approximate which foods could be more important, such as crops capable of sustaining reasonably high yields with low temperatures and high water stress, or food sources independent of these conditions. The catastrophe strikes in May, which is likely a worst-case scenario. Models suggest that the climactic effect of particle emissions is larger during this period; additionally, food stocks are at their near lowest in the global North, as the Northern hemisphere harvest season has not yet started, and could potentially fail [38]. The results found using this model are also, to some extent, applicable to the volcanic winter scenario [22,37,39], even though the soot aerosols from nuclear fires result in more extreme climate effects.

### 2.3. Resilient Foods

Resilient foods were selected upon their expected availability in the event of an ASRS. They include traditional foods such as cool-tolerant staple crops that could be grown outside, or other major staple crops that could be grown in greenhouses, but also innovative resilient foods that are scalable and could provide a significant portion of the world’s dietary needs. Affordability and caloric availability were also taken into consideration to exclude some potential food sources, such as mushrooms, insects, canola meal, fresh fruits and vegetables, nuts, synthetic fat, leaf protein concentrate and others; these are discussed in Appendix B.

#### 2.3.1. Traditional Resilient Foods

##### Potato

Potatoes are cool tolerant [40,41,42] and have been proposed as a food resilient to global cooling [43], thus it was hypothesized that their production will still be possible in the proposed sunlight reduction scenario, especially after the first year has passed, if crop relocation has been undertaken. The nutritional analysis used boiled potato flesh, cooked in skin without salt. The intake was limited to 500 kcal (575 g) per day per person to reflect limited availability, but larger intakes are safe.

##### Wheat, Barley, Canola

Some staple crops, namely wheat, barley and canola (a variety of rapeseed; for canola oil), could be eligible for outdoor growing, if relevant adjustments are made to relocate them to appropriate regions. In the first months after the catastrophic event, significant amounts of these would be available as food stocks. Representative examples of foods produced from these crops were used in the nutritional analysis. For wheat, white wheat flour was used in period I, while hard red spring wheat was used in periods II and III. This cultivar appears to be a good proxy for what could be grown in significant quantities in tropical latitudes in the considered scenario. Moreover, cooked pearled barley and oil from canola seeds were also used.

##### Rice, Maize, Soybeans

Rice, maize, and soybeans are three staple crops that would be highly demanded, as they constitute a significant part of the global population’s diets—representing respectively 13, 12 and 3% of global crop production in 2019, by weight [44]—, and the latter being rich in proteins and fats. It is hypothesized that they could potentially be grown in greenhouses, as they are warm-environment crops. They could be available in stocks first, and then via greenhouse harvesting. Scaling up of low-cost open source greenhouse agriculture during a food GCR scenario has been described elsewhere [25]. Previous research suggests that it would take between 6 months and 1 year to build a significant number of greenhouses and start growing crops. The foods used here were cooked rice (white, long-grain, regular, unenriched, in period I; and brown, long-grain in periods II and III); cooked sweet yellow corn and corn flour; and soy flour (full-fat) and mature, cooked soybeans.

##### Fish

Fishing would presumably still be possible in the first year after the catastrophe [45]—although yields would be reduced compared to current values [37,45]. Anchovies were integrated into the model, as it is the top fished species (*Engraulis ringens*) with 7.0 million tons caught in 2018—10% of the global marine capture [46]. It is foreseen that there would be some availability during the first year that will decrease over time during the rest of the catastrophe, due to a reduction in their food sources from lower photosynthetic capacity in low sunlight scenarios. Raw European anchovy (*Engraulis encrasicholus*) nutritional values were used as a proxy for any fish that might be consumed.

##### Meat and Animal Organs

As food will be scarce, a much higher number of animals than usual could be slaughtered at the beginning of the catastrophe due to lack of resources to feed them, as animal farming is a low-efficiency food conversion method (the caloric conversion efficiency typically ranges between 3–17%) [47]. A significant number of ruminants, however, could be primarily fed non-human-edible material, such as cellulose-rich agricultural residues [15,48], though investing these resources on milk rather than meat production would be ~6 times more efficient on a caloric basis [47]. Non-ruminating animals were not included here as they feed primarily on human-edible resources and would thus act as competitors rather than primary food sources. The animals that could not be fed this way would likely be slaughtered for food and used for both their meat and their organs, making these foods highly available during the first few months. After that, meat and organs would likely become scarce and the costs would climb dramatically due to limited means for feeding the animals. The foods used in the analysis here were lean meat, fat, and organ composite from cattle, calculated using the relative proportions of each organ in cattle as well as their specific nutrient content.

##### Milk

Maintenance of some amount of dairy livestock could be crucial as milk could be one of the few sources of fats in the diet, especially during the first year, as oil-producing plants are not guaranteed to be available in very significant quantities. Again, a significant number of dairy livestock could be maintained on a diet primarily based on agricultural residues [48]. Milk production might be available for an extended period if sufficient grazing areas and sustained generation of cellulosic residues are present, but probably in limited quantities. Unfortified whole milk with 3.25% milkfat was used.

##### Sugar Beet

Sugar beets have been proposed as a food resilient to global cooling [43], and might be feasible to grow outdoors in some areas even during an ASRS, as they tolerate lower temperatures and sunlight levels than cereal crops. They could potentially make a significant contribution to people’s energy uptake, thus helping to bridge a possible caloric gap.

#### 2.3.2. Alternative Resilient Foods

##### Seaweed (Macroalgae)

Seaweed can grow with low sunlight [49] and would be protected from ultraviolet light possibly generated by a nuclear winter [50]. Developing seaweed farming has been proposed as a potentially very significant contribution to the dietary intakes of the global population in this situation [38]. Seaweed consumption has been advocated by the FAO for decades [51,52,53]. It is fast-growing and highly scalable [54,55], and seaweed aquaculture does not compete with terrestrial agriculture for fresh water, land, or pesticides [56]. In 2012, seaweed harvests accounted for 3 million dry tonnes [54], so there is already substantial demand and cultural acceptance as a food in many regions of the globe. With significant up-scaling effort, substantial seaweed production could be possible as soon as 6 months after the disaster [38]. The seaweed used were emi-tsunomata, laver, and wakame.

##### Single-Cell Protein from Bacteria

Certain single-cell organisms—some species of bacteria, microalgae, and fungi—can be leveraged for production of protein-rich foods. Single-cell protein from agriculture-independent sources such as methane- or hydrogen-oxidizing bacteria could complement or replace animal proteins [57]. Ramping up this technology has the potential to make a significant contribution to fulfilling the protein requirements of the global population [26,28]. Here it is estimated that it would be available around one year after the catastrophe. Nutritional data from the methane-oxidizing bacteria of Unibio A/S was used as a proxy [58]. Spirulina (*Arthrospira platensis*) was used as well as a representative example of the group of cyanobacteria sometimes considered to be microalgae, including spirulina, chlorella, and others.

##### Lignocellulosic Sugar

Quick deployment of technologies to produce sugar from lignocellulosic biomass could make an important contribution to fulfilling the caloric requirements of the global population during the catastrophe period. A previous analysis suggested this food source could be ramped up to a significant amount 6 months after the onset of the catastrophe [59]. A typical Atwater factor of 4 kcal/g was used [60].

### 2.4. Investigation of Suitable Diets

#### 2.4.1. Chronology of the Resilient Foods’ Availability

A chronology of the availability of these resilient foods in the context of the proposed ASRS is presented in Table 1. The time following the catastrophe was divided into three periods, each expected to last between 3 and 9 months. There remains significant uncertainty in the timing of the potential availability of the proposed food sources, but overlapping availability could be sufficient to consider combining the associated food sources in post-catastrophe diets. Food in Period I (food stocks-vegetable and animal), following the catastrophe, would mainly be from previously stored stocks of major staple crops (those typically used to provide a dominant portion of many human diets globally, such as wheat, barley, rice, canola, maize and soybeans). Animal stocks would also be available and might be slaughtered in great numbers due to plummeting feed availability. Meat, organs, fish and milk are presumed available. Potatoes might still be grown in the scenario considered, and sugar should be available with sugar beet crops. After stocks are nearly or completely depleted—between 3 and 6 months after the shock [15,61]—and a significant share of existing livestock has been slaughtered globally, Period II is projected to start, where diets might be made of mainly potatoes, with marginal contributions of the major staple crops’ stocks and harvests (from outdoor and greenhouse growing). White rice and wheat flour were replaced by brown rice—that needs less processing—and spring wheat—that should be easier to grow under the new climatic conditions, respectively. Other available foods are animal products (meat, organ, milk, fish), and seaweed [38]. Lignocellulosic sugar and seaweed could technically undergo significant production ramp-up during this period, potentially making a substantial contribution to the global food requirement [38,59]. A few months after that, if crop relocation and/or greenhouse ramp-up [25] are successful, a significant amount of staple crops could become available again, and possibly significant quantities of single cell protein as well, if industrial production ramp-up succeeds [26,28]. This would be Period III, which might extend longer than the two first periods. Other potential resilient foods, which might also be available, are discussed in Appendix B.

#### 2.4.2. Nutrient Analysis

Table 2 provides the nutrients selected for analysis. Nutritional values for all of the resilient foods listed in Table 1 were obtained mainly from Food Data Central [62] and complemented by various academic literature sources (see Appendix A).

Diets combining foods available simultaneously—according to the proposed chronology—were created. Relative proportions of foods were tuned to meet the 2100 kcal energy intake and as many AIs as possible, or at least MRAIs.

## 3. Results and Discussion

### 3.1. Proposed Diet Combinations Based on Resilient Foods

In this Section, some feasible diets are presented for the different periods defined in Section 2.4.1. The foods expected to be available simultaneously were combined and the resulting nutrient profile was analyzed, with a focus on potential deficiencies. For each period, two diets were proposed: one in an optimistic scenario, where all the expected resilient foods are effectively available and can be combined together; and a pessimistic one where one key resilient food source is missing. The proposed food combinations are presented in Table 3, while the nutrient profile of each combination is shown in Figure 2. Amino acid profiles were not presented here, as none of the diets presented any amino acid deficiencies. Further discussion is provided in Appendix C (Table A1 and Table A2).

The words ‘moderate(ly)’ and ‘severe(ly)’ are used to refer to MRAIs and SRAIs.

#### 3.1.1. Period I

##### Optimistic Scenario

In an optimistic scenario for Period I, the included foods come from cattle (meat, organs, milk), fish, as well as staple crops (as stocks: rice, barley, canola, soy, wheat) and potatoes (Table 3). Many of the nutrient AIs were met (Figure 2), with deficiencies in vitamins D, E, C, and K, particularly D and E (SRAIs). Calcium intake was moderately low, but within WHO recommendations during emergencies.

##### Pessimistic Scenario—Without Staple Crops’ Stocks

In a pessimistic scenario for Period I, it was proposed that stocks of staple crops (wheat, barley, canola, rice, corn, and soy) would be unavailable and that most people would feed on potatoes, fish, cattle, and sugar (Table 3). The diet proposed is, in this case, deficient in several micronutrients (Figure 2). Essential fatty acid intake was severely low (alpha-linolenic acid (ALA) and linoleic acid (LA)), which could be palliated by bacteria and microalgae-based foods. Fiber intake was severely low as well, but this could be countered by consuming whole grains, skins, and husks. Protein intakes appear dangerously high. Vitamins D, E, and K are again problematic, as well as manganese, magnesium, and, to a lesser extent, potassium.

#### 3.1.2. Period II

##### Optimistic Scenario

It is envisioned that during an optimistic Period II, people could still feed on potatoes, major staple crops (wheat, barley, canola, rice, corn, and soybeans), fish, and animals, but at lower levels than Period I (Table 3). It means that stocks are not completely depleted and/or harvesting was rendered possible by relocating crops and/or building greenhouses. Seaweed is expected to start making a significant contribution to the diet. The nutrient intake seemed adequate, with only one SRAI—vitamin D (Figure 2).

##### Pessimistic Scenario—Without Seaweed

The pessimistic diet that was investigated for Period II was if seaweed had not been scaled up sufficiently to be proposed (Table 3). As a consequence, overall food availability could be reduced and feeding everyone might prove even more difficult. They have been replaced in their entirety by an intake of lignocellulosic sugar, to make up for the missing calories. The nutrient profile is similar to the optimistic scenario, except for mineral intakes (Figure 2). Again, vitamins D, E, C, and K are missing, with the first two reaching SRAI levels. Calcium intake is also low.

#### 3.1.3. Period III

##### Optimistic Scenario

During Period III, it is projected that staple crops would be available in higher quantities, but fish and animal meat and organs would be lacking. Moreover, seaweed would be available, as well as single-cell protein (here from methane-oxidizing bacteria), and optimistically milk from grazing cattle too (Table 3). In the proposed diet, two nutrients reached SRAIs: vitamin D and manganese (Figure 2). Vitamins E and K and calcium are again moderately deficient, and vitamin A becomes deficient in this period compared to the others due to lower availability of animal-sourced foods (e.g., cattle organs and milk).

##### Pessimistic Scenario—Without Greenhouses

For Period III, the pessimistic scenario assumes a failure of ramping up greenhouse production sufficiently (Table 3). Hence, rice, maize, and soybeans are not available. The nutrient profile is quite similar to the optimistic scenario, except that some LA and vitamin K are severely lacking (Figure 2).

### 3.2. Risks Associated with Inadequate Intakes

#### 3.2.1. Macronutrients

##### Proteins

In all proposed diet combinations, proteins are generally in excess, mostly due to the high availability of meat in the first two periods of the ASRS from the presumed slaughter of most farmed animals due to lack of feed. The MRAI upper limit was set to 140 g per WHO recommendations [34]; however, Bilsborough and Mann reported that obtaining over 35% of the energy intake as proteins (184 g for 2100 kcal) is ‘dangerously excessive’ while 25% (131 g) is safe [63], hence, while this does not appear to be risk-free, it is likely tolerable for a certain amount of time, especially during a catastrophe.

##### Fats and Fatty Acids

Omega 6 fatty acids (LA) are insufficient in all diets, except for the optimistic one in Period I. It reaches SRAI in Period III if greenhouses are not available. Severe deficiency has been linked to dermatitis in animal models [64], and imbalances in the ratio of omega 3 and omega 6 fatty acids have been linked to atopic, allergic, visual, attentional, emotional, and sleep problems [65]. Vegetable oils (e.g., soybean, corn, canola) are major omega 6 sources that could palliate this issue.

##### Fiber

Low fiber intake, observed in Period I pessimistic scenario (13 g), is associated with constipation and digestive tract disorders. Interestingly, the average intake of fiber is 15 g/day for American adults [66], while 25 g is recommended [31]. It is concluded that this issue would be tolerable for at least a short amount of time, e.g., one period of 3 to 9 months, and it could be alleviated by consuming whole grains, skins, and husks.

#### 3.2.2. Micronutrients

Potential micronutrient deficiencies based on the proposed diets are presented in Table 4, as well as strategies to mitigate them. Further discussion is provided in Appendix D.

The current analysis highlights the difficulty of providing an adequate supply of some micronutrients, markedly vitamins D and E. Deficiencies of these vitamins, however, are largely common across vast portions of the global population today [72,73,74], indicating that population survivability may not be hampered during the catastrophe even if these deficiencies were present. More research on efficient, adequate interventions such as fortification and supplementation schemes is useful for both addressing this problem today and during a catastrophe. Strategic micronutrient supplement stockpiles could be a useful resource for addressing this problem during global food catastrophes.

Some micronutrients were in excess. Niacin is often in excess but is associated with low toxicity—mainly flushing of the face, which is a mild effect [75]. Vitamin B12 was in excess in Period I, but it has not been linked to significant side effects [75]. Moreover, as the availability of animal foods will decline with time, body storage of vitamin B12 for future physiological needs might be a way of preventing later deficiencies. Vitamin A was in excess in Period I, which is associated with various disorders depending on the actual source of the vitamin [76]. Copper was usually in excess; this might affect antioxidant status and immune function [77]. Manganese was slightly in excess in Period III, but studies show no evidence of toxicity from levels higher than these [78]. Iodine was also in excess in Period III, mostly due to seaweed intake, but boiling the seaweed can reduce iodine content significantly [79].

### 3.3. Suitability of Each Diet and Comparisons among Periods

It appears that adequate diets can be provided during Period I in an optimistic scenario. In this scenario, only vitamins D and E were severely lacking and means of mitigation were discussed in Section 3.2.2. In the pessimistic scenario, however, many deficiencies are present, and missing calories were replaced by potentially harmful amounts of proteins. Significant efforts should be made to carefully manage staple crops’ stocks in order to avoid global famine and malnutrition and more effort is needed in identifying resilient sources of balanced nutrition.

For Period II, the optimistic scenario seemed to be nutritionally adequate. However, some chronic deficiencies, e.g., in vitamin D, might appear. The pessimistic scenario seemed relatively complete too, but it should be noted that, even though the nutritional balance appears adequate, food availability would be lower in this scenario, hence increasing famine and malnutrition incidence. Chronic deficiencies might kick in during this period if some nutrients cannot be provided over several periods.

Fairly balanced diets were found for Period III, provided that people manage to make up for potential chronic deficiencies—e.g., in vitamins D, E, K, omega 6, maybe calcium—and that potential new deficiencies—vitamins A and B12—are dealt with in time. In the optimistic scenario, manganese excess should be handled too. Here again, it should be pointed out that the pessimistic scenario diet, as nutritionally adequate as it might seem, has the flaw of not being able to feed as many people as the optimistic scenario diet should.

It is worth restating that these prospective diet combinations are not considered optimal nor exclusive. They simply aim to represent combinations of reasonable amounts of food sources thought to be available simultaneously during an abrupt sunlight reduction catastrophe based on the current state of the research while attempting to fulfill as many essential nutrient requirements as possible, especially the nutrients whose deficiencies are linked to severe health disorders. Further investigation and optimization of these diets would be a significant contribution to ASRS preparedness. For example, methods such as gradient descent over multi-dimensional weighted distance-to-optimal-intake function or genetic algorithms could be useful. These could be incorporated into future scenario analyses for ASRS response integrating different resilient foods to study nutritional considerations globally or regionally.

### 3.4. Uncertainties and Mitigation

#### 3.4.1. Catastrophe Scenario Considered

A scenario of 150 Tg of soot injected into the upper troposphere and stratosphere during a nuclear war was used as a basis, which corresponds to only one possible outcome of a range of potential sunlight reduction scenarios. Given the absence of empirical data on the severity of a nuclear winter scenario, works like this must rely on climate models of post-catastrophic climate [22,37] to infer the global dietary resources and needs, with the inherent uncertainty this brings. Other sources of uncertainty, related to the severity of the scenario, include the number of target cities, the amount of smoke produced from this, how much of the soot ends up in the atmosphere, the degree to which these would alter temperature, light, and precipitation levels and the duration of the effects [80]. Similar concerns apply to the other ASRS catastrophes.

Other significant uncertainties in this study include the degree to which global industry and energy, trade and international relationships, etc. are affected. During the COVID-19 pandemic, substantial disruptions have been identified [81,82,83,84]. There can also be counterproductive policy [85]. Uncertainties may vary with, e.g., the nature of the abrupt sunlight reduction shock (volcanic winter, nuclear winter, asteroid/comet impact, or an unknown scenario not covered in literature), or for the nuclear war scenario: involved countries, duration and extent of the conflict, weapons used, but also remaining diplomatic relationships and trade, etc. In the analysis shown here, international cooperation and trade were assumed. Global industrial and energy producing infrastructure were considered to remain largely unaffected. In a more extreme scenario in which these are disabled in combination with the ASRS, a different food solutions portfolio would have to be proposed [86].

Regarding the severity of ionizing radiation, the internal dose (which is mostly from food) would be negligible compared with the external dose following a major nuclear war [87]. There is still live discussion in the scientific literature about cancer deaths from the long-term effects of radiation in Hiroshima and Nagasaki. Sutou states that people who lived in Hiroshima and Nagasaki areas after the atomic bombing did not have a shorter life expectancy than the general Japanese population [88]. Other articles estimated cancer deaths from 430 [89] to 1900 [90]. For comparison, deaths from the immediate effects of the atomic bombs in Hiroshima and Nagasaki were around 200,000.

Nonetheless, the analysis is applicable to any scenario involving a catastrophic abrupt sunlight reduction, whether it happens following a nuclear war, a large volcanic eruption, a very large asteroid or comet impact.

#### 3.4.2. Nutritional Values

There is no guarantee that the nutritional needs of the population will remain the same in the event of an abrupt sunlight reduction catastrophe as described here, nor that, if they were to remain unchanged, the selected values are applicable across continents and cultures. Additionally, increasing prevalence of malnutrition and disease in such a starvation-prone period might affect digestibility and bioavailability of several nutrients [91], potentially influencing the suitability of the diet combinations. Moreover, there is uncertainty regarding whether the nutritional values of the resilient foods would remain the same; reduced sunlight and temperatures could significantly affect the nutrient content of crops and is an area in need of further research. For instance, immature grains could be more nutrient-dense, but less calorie-dense than mature grains, as is the case for rice [92]. This would affect the overall nutrient profile of the diets. Lower sunlight and temperatures may not be sufficient for the crops to reach maturity. Moreover, lower ultraviolet irradiation could reduce the vitamin D content of foods. Whether this takes place would depend on the nature and severity of the scenario.

Moreover, for the sake of simplicity, this analysis tackles only a subset of adequate nutrition, namely 2100 kcal for the global average 62 kg [24] of body weight. However, there is variability in nutritional needs across ages, genders and conditions, e.g. it would be insufficient for most adult men. Many subgroups (e.g., infants, children, pregnant or lactating women) should be considered in future research, as they have different nutritional needs [29,31,34]. This choice and its appropriateness are further discussed in Appendix E.

It is worth pointing out that nutrition science and recommendations are greatly debated in the scientific community as they are often based on surveys rather than more accurate, but more difficult to obtain physiological data [93,94,95]. Adding this to the uncertainties around nutrition in a post-catastrophic situation, caution is recommended in handling the results of this work, as they are prone to revision due to changes and improvements in the scientific understanding of nutrition, and the understanding of the potential of resilient foods, which ought to be addressed in future work. Moreover, billions of people worldwide do not have healthy—or even nutritionally adequate—diets according to the recommendations used here [29,30,31,32,33,34,35], as reported by Our World in Data [96]. Hence, it is difficult to assess how severe some deficiencies could be. Additionally, while regional or cultural acceptance of foods such as seaweed or meat might be limiting, it should be noted that the primary determinants of food choice in extreme situations are perceived nutritional value and feeling of fullness provided by the foods [97].

Uncertainties concerning protein digestibility are discussed in Appendix E, as well as the role of antinutrients—substances that can interfere with nutrient absorption through various mechanisms—and contaminants.

#### 3.4.3. Availability of Resilient Foods

It is worth restating that the availability of different foods over time during an ASRS as proposed in this work represents a reasonable and technically feasible scenario in which quick response and cooperation allow humanity to avoid mass global starvation. This cooperation needs to be centered not only in trade between areas that can produce resilient foods and those that cannot but also in terms of making information and technologies openly available so that all those that have the potential to produce resilient foods have the adequate knowledge to do it. There is no guarantee, however, that such cooperation will effectively be available if such an event occurs, as there are many uncertainties concerning how such a scenario would develop. Many variables can impact these conclusions, such as the extent of the sunlight decrease (which could force relocation to lower latitudes and therefore changed photoperiods, rendering some crops impossible to grow, though a small amount of supplemental artificial light could ameliorate this [25]), the global and regional temperature decreases (that would impact transport distances and regional food availabilities) and the degree of cooperation for industry up-scaling (that could impact single cell protein, cellulosic sugar, and/or greenhouse production). Another uncertainty is the degree of information and willingness of the population to use distributed food production techniques to meet their needs as in the case of the COVID-19 pandemic [98,99]. It should also be pointed out that the selected timing of the catastrophe is only one possible scenario; a different timing would likely imply a somewhat higher food availability at the onset, due to different food stock levels, degree of aerosol lofting, and harvest timing.

Chronological analysis was performed on periods of several months—rather than month-by-month—making results less precise but more robust to potential changes in the knowledge of an ASRS, such as the suitability and potential of the foods proposed. Some uncertainty still remains on whether the availability of different resilient foods would overlap in this way, and on whether the foods would be available in sufficient quantities. These will be addressed in future work. Moreover, this analysis included a somewhat large variety of foods, since dietary diversity is a key element of an adequate nutrition intake [100]; if a lower amount of food sources would prove to be available, the outcome of this study would be worse, as individual nutrient requirements could be more difficult to fulfill.

#### 3.4.4. Other Resilient Foods

There are several other classes of resilient foods that could bring useful nutritional contributions in these contexts that were not included in the diet combinations analyzed. They are discussed in Appendix B.

### 3.5. Future Work

The dietary combinations proposed in this study are not prescriptions, rather they serve to determine how feasible a nutritionally balanced diet is in the proposed scenario, based on low-cost foods that would be available provided there is sufficient preparedness and a fairly quick and coordinated response. Finding out in advance how the foods likely to be available to the most vulnerable in these most extreme scenarios can be adequately combined could help prevent mass starvation. This would be an important step not only towards fulfilling the ‘Zero Hunger’ sustainable development goal [101], but also towards achieving the more ambitious ideal of feeding everyone no matter what.

The overarching goal of this line of research is to strengthen resilience and response for food-related GCRs. Resilience and response strengthening have been proposed as important facets of a robust defense against existential risk as well [102] as they could help reduce existential risk factors, of which social unrest due to global famine could be an example [103]. Preparing for worst-case scenario climatic catastrophes could also potentially inform food researchers and policymakers on how to reduce the death toll of food shocks, such as MBBF, that are quite severe, but that do not risk endangering the survival or future of humanity.

The current model is based on partially validated conclusions of ongoing research on the topic of catastrophe-resilient foods. More work is needed in several areas: on the one hand, nutritional values need to be assessed with more confidence and precision, especially for application in a catastrophe scenario; on the other hand, reducing uncertainty in the potential availability of foods is paramount, perhaps via complete integration of climate models and regional analyses based on Geographic Information System (GIS) methods to assess the situation on a regional rather than global basis. GIS-based analysis can also be helpful for identifying targets for research in resilient foods [104]. Better understanding of micronutrient deficiency mitigation pathways, such as industrial production ramp-up, supplementation and fortification schemes, can also greatly improve preparedness in food-related GCRs.

As previously mentioned, this analysis is focused mainly on the technical feasibility of providing adequate nutrition during the most extreme global food catastrophes. Including more economic considerations and modeling would improve the understanding of the food landscape after a food catastrophe originating from an abrupt sunlight reduction. Finally, understanding some other, more uncertain factors, such as the degree of post-catastrophe coordination between regions, the degree of government intervention in food systems, such as via price ceilings, rationing or food export bans, could help further strengthen catastrophe preparedness. This work aims to serve as a basis for the nutritional study of future scenario analyses incorporating the considerations discussed in this section.

By combining the methodology presented in this study with analyses of specific catastrophic food scenarios based on concrete sets of assumptions, the specific forms of malnutrition that can be expected will be highlighted, as well as specific foods and interventions that could be implemented. Upcoming research will integrate this type of nutritional analysis with a parallel work analyzing food production, availability, and affordability in specific ASRS scenarios.

## 4. Conclusions

The aim of this work is to pave the way towards a better understanding of the nutrition landscape in an abrupt sunlight reduction scenario (ASRS), such as a nuclear or volcanic winter. A nuclear winter scenario as proposed in the literature is used as a model ASRS to study how even an effective response to this extreme food crisis would likely result in nutritional insufficiencies, what these are likely to be, and how to mitigate them. While many uncertainties remain regarding the scenario of a sudden decrease in sunlight reaching the surface of the Earth, the role of planning and preparedness in the outcome of such a catastrophe is clearly vital. Indeed, the subsequent collapse of agriculture would drive humanity into starvation if proper preparation were absent.

This study analyzes potential combinations of resilient foods into feasible, balanced diets that would help mitigate the impact of this catastrophic scenario given an effective response to it, including rapid food production ramp-up. A list of low-cost resilient foods based on current technologies is proposed, from which nutritionally adequate diets appear technically achievable, even for the most at-risk populations in the most extreme food catastrophes. No life-threatening nutrient deficiencies or excesses were found, meaning that given sufficient availability of resilient foods, the worst consequences of malnutrition appear avoidable. However, some micronutrients were found to be potentially deficient over long periods of time, e.g., vitamin D and E. Careful management and cooperation over our food sources—storage, trade, production ramp-up, etc.—would be essential to ensure food availability and prevent the worst outcomes. Adequate preparedness is indispensable in achieving the goal of feeding everyone no matter what. This study aims to serve as a basis for the nutritional study of scenario analyses during future research.

## Figures and Tables

**Figure 1 nutrients-14-00492-f001:**
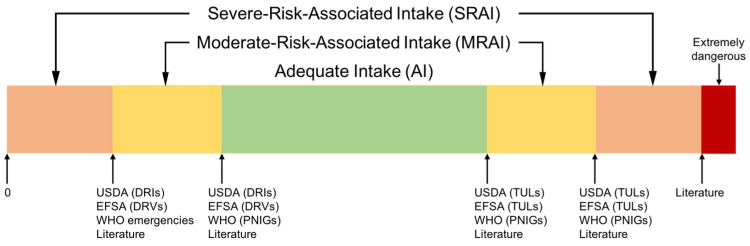
The three proposed intake categories: adequate intake (AI), moderate- and severe-risk-associated intakes (MRAI and SRAI). Threshold values were found in USDA DRIs and TULs [29,30], EFSA DRVs [31,32] and TULs [33], WHO PNIGs [34,35] and emergency management handbooks [23,36], and in other literature (see Appendix A).

**Figure 2 nutrients-14-00492-f002:**
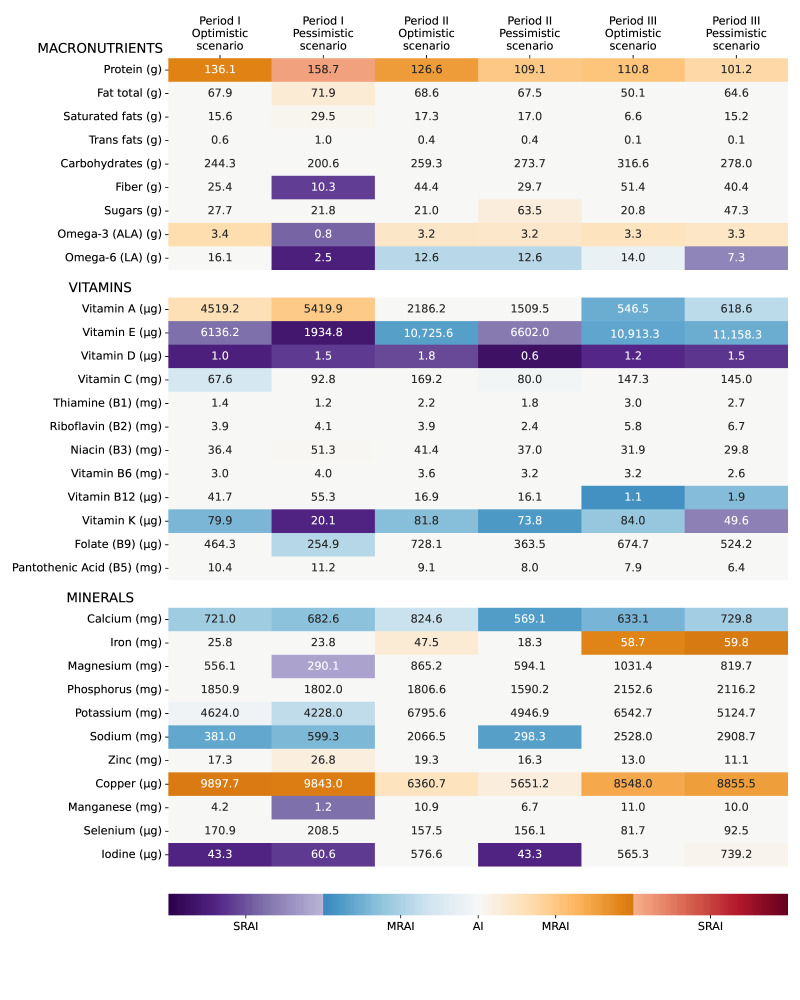
Nutrient distribution in optimistic and pessimistic scenarios for each period.

**Table 1 nutrients-14-00492-t001:** Proposed chronology of resilient foods’ availability—in red: not available in significant quantities; in yellow: possibly available in significant quantities; in green: available. Each period is expected to last between 3 and 9 months.

Food Source	Period I: Food Stocks (Vegetable and Animal)	Period II: Potatoes and Resilient Food Ramp-Up	Period III: Greenhouses, Crop Relocation and Alternative Resilient Foods
Potato			
Wheat, Barley, Canola	Stocks		Relocation
Maize/Corn, Rice, Soybeans	Stocks		Greenhouses
Fish			
Meat/Organs			
Milk			
Sugar	Beets	Lignocellulosic Sugar	Lignocellulosic Sugar
Seaweed			
Single-Cell Protein			

**Table 2 nutrients-14-00492-t002:** Nutrients selected for analysis of foods and for which AIs, MRAIs and SRAIs were determined (see Appendix A).

Macronutrients					Micronutrients	
Proteins		Fats		Carbohy-Drates	Minerals	Vitamins
9 Essential Amino Acids: Histidine, Isoleucine, Leucine, Lysine, Methionine, Phenylalanine, Threonine, Tryptophan, Valine	2 Derived Amino Acids: Cysteine, Tyrosine ^a^	2 Essential Fatty Acids: PUFA 18:2n-6 (Linoleic Acid (LA), An Omega-6) and 18:3n-3 (Alpha-Linolenic Acid (ALA), an Omega-3) ^b^	Saturated Fats, Trans Fats	Sugars, Fiber	Calcium (Ca), Iron (Fe), Magnesium (Mg), Phosphorus (P), Potassium (K), Sodium (Na), Zinc (Zn), Copper (Cu), Manganese (Mn), Selenium (Se), Iodine (I)	Vitamins A, E, D, C, B6, B12, K, Thiamin (B1), Riboflavin (B2), Niacin (B3), Folate (B9), Pantothenic Acid (B5)

^a^ Cysteine is derived from methionine, tyrosine from phenylalanine. ^b^ Where unavailable, values of undifferentiated 18:2 and 18:3 PUFA were used as proxies.

**Table 3 nutrients-14-00492-t003:** Distribution of caloric and weight intake of resilient foods in optimistic and pessimistic scenarios in each period.

Period	Period I—Food Stocks (Vegetable and Animal)				Period II—Potatoes and Resilient Food Ramp-up				Period III—Greenhouses, Crop Relocation and Alternative Resilient Foods			
Scenario	Optimistic Scenario		Pessimistic Scenario		Optimistic Scenario		Pessimistic Scenario		Optimistic Scenario		Pessimistic Scenario	
Unit	g	kcal	g	kcal	g	kcal	g	kcal	g	kcal	g	kcal
Potatoes	344.8	300	574.7	500	574.7	500	574.7	500	574.7	500	574.7	500
Wheat Flour	68.7	250	-	-	-	-	-	-	-	-	-	-
Wheat (Hard Red Spring)	-	-	-	-	85.1	280	85.1	280	76	250	106.4	350
Barley (Pearled)	24.4	30	-	-	24.4	30	24.4	30	40.7	50	40.7	50
Canola Oil	19.2	170	-	-	20.4	180	20.4	180	22.6	200	28.3	250
Rice (White)	115.4	150	-	-	-	-	-	-	-	-	-	-
Rice (Brown)	-	-	-	-	81.3	100	81.3	100	81.3	100	-	-
Corn Flour (Whole-Grain)	27.7	100	-	-	-	-	-	-	41.6	150	-	-
Corn	104.2	100	-	-	-	-	-	-	140.6	135	-	-
Soy Flour	46.1	200	-	-	51.8	225	51.8	225	46.1	200	-	-
Soybeans	116.3	200	-	-	29.1	50	29.1	50	46.5	80	-	-
Anchovy (Raw)	76.3	100	114.5	150	76.3	100	76.3	100	-	-	-	-
Cattle (Lean)	49.3	100	221.7	450	73.9	150	73.9	150	-	-	-	-
Cattle (Fat)	-	-	36.8	250	16.2	110	16.2	110	-	-	29.4	200
Cattle (Organs)	178.8	250	214.6	300	57.2	80	57.2	80	-	-	-	-
Milk (Whole)	245.9	150	327.9	200	204.9	125	204.9	125	41	25	163.9	100
Sugar (Beets)	-	-	64.9	250	-	-	-	-	-	-	-	-
Spirulina (Dry)	-	-	-	-	-	-	-	-	17.2	50	17.2	50
Emi-Tsunomata (Dry)	-	-	-	-	38.6	100	-	-	38.6	100	38.6	100
Laver (Dry)	-	-	-	-	33.3	70	-	-	21.4	45	23.8	50
Wakame (Dry)	-	-	-	-	-	-	-	-	5.6	15	9.3	25
Bacteria (Methane)	-	-	-	-	-	-	-	-	38	200	57	300
Lignocellulosic Sugar	-	-	-	-	-	-	42.5	170	-	-	31.3	125

**Table 4 nutrients-14-00492-t004:** Micronutrient deficiencies that could be present in diets resilient to ASRS and selected potential sources and mitigation strategies.

Micronutrient	Reason for Inclusion	Sources Available in a Sunlight Reduction Scenario	Notes
Vitamin D	Deficiency in all periods and considered by some to be widespread in all age groups in current times [67].	Sunlight exposure, UV-treated biological sources (mushrooms, yeast, lichen), fatty fish (mackerel, salmon, sardines), dairy, kelp.	The current standard supplement production method is based on sheep wool, which may not be widely available in an ASRS, if animal agriculture is significantly reduced as expected. The sunlight exposure method could be insufficient for an ASRS in which UV is reduced, and dangerous in an ASRS in which UV is increased due to ozone layer destruction.
Vitamin E	Deficiency in all periods.	Vegetable oil (canola, soybean, corn), fortification/supplementation (industrial chemical synthesis [68]).	Current chemical synthesis capacity could cover a significant part of the minimum recommended requirements of the global population (see Table A3). Certain types of leaf protein concentrate could be a good source (see Appendix B).
Vitamin K	Deficiency in all periods.	Canola oil, some seaweeds, fermented foods (soybeans, etc.).	Fermenting soybeans can increase their vitamin K2 content ~100-fold [69]. Certain types of leaf protein concentrate could be a good source (see Appendix B).
Vitamin A	Moderate deficiency in period III, also high severity and global prevalence in current times [67].	Dairy, liver, fish, fortification/supplementation (industrial chemical synthesis [68]).	Current global production via industrial synthesis appears sufficient to fulfill the minimum recommended requirements of the global population (see Table A3).
Vitamin B12	Moderate deficiency in period III.	Meat, organs, fish, single-cell protein from methane- or hydrogen-oxidizing bacteria, fortification / supplementation (industrial biochemical synthesis [70]).	Current global production via industrial synthesis appears sufficient to fulfill the minimum recommended requirements of the global population (see Table A3).
Iodine	Deficiency in period I, also high severity and global prevalence in current times [67].	Seaweed, fish, shellfish, dairy, fortification/supplementation (mining).	Commonly used in fortification in the form of iodized salt. Seaweed is an excellent source which appears very suitable to avoid this deficiency in an ASRS. Additionally, current global production of iodine appears more than sufficient to fulfill the minimum recommended requirements of the global population.
Calcium	Deficiency in all periods and estimated to be widespread now [67].	Dairy, fish and land animal bones, fortification/supplementation (mining, bone meal).	Current global production of calcium carbonate alone appears more than sufficient to fulfill the minimum recommended requirements of the global population.
Vitamin C	Common in famines and displaced persons, high severity [67].	More potatoes, fortification/supplementation (industrial chemical synthesis [68]), some seaweeds.	Current global production via industrial synthesis appears sufficient to fulfill the minimum recommended requirements of the global population. Certain types of leaf protein concentrate could be a good source (see Appendix B).
Thiamine (B1)	Common in famines and displaced persons, high severity [67].	Single-cell protein from methane- or hydrogen-oxidizing bacteria, fortification/supplementation (industrial chemical synthesis [68])	Current global production via industrial synthesis appears sufficient to fulfill the minimum recommended requirements of the global population.
Niacin (B3)	Common in famines and displaced persons, high severity [67].	Potatoes, whole wheat, barley, brown rice, mushrooms, fish, meat, organs, dairy, single-cell protein from methane- or hydrogen-oxidizing bacteria [71].	
Vitamin B6	Common in famines and displaced persons, high severity [67].	Fortification/supplementation (industrial chemical synthesis [68])	Current chemical synthesis capacity could cover a significant part of the minimum recommended requirements of the global population.
Iron	High severity and global prevalence in current times [67].	Meat, organs, fortification/supplementation (mining), single-cell protein from methane- or hydrogen-oxidizing bacteria [71].	Current global production of ferrous sulphate only is more than sufficient to fulfill the minimum recommended requirements of the global population.
Zinc	High severity and global prevalence in current times [67].	Fortification/supplementation (mining), meat, organs, shellfish, dairy, single-cell protein from methane- or hydrogen-oxidizing bacteria [71].	Current global production of zinc sulphate only is sufficient to fulfill the minimum recommended requirements of the global population.

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
