# Peer review of "Nutrition in Abrupt Sunlight Reduction Scenarios: Envisioning Feasible Balanced Diets on Resilient Foods"

_nutrients, 2022, doi:10.3390/nu14030492_

Round 1

Reviewer 1 Report

The article entitled Nutrition in abrupt sunlight reduction scenarios: envisioning feasible balanced diets on resilient foods submitted for evaluation was written correctly and shows that a considerable effort went into your work. 

Some comments/recommendations:

  1. Is recommended to clarify what is (and why) resilient foods in the context of the study.
  2. The model ASRS used in the study should be better explained in the "methods" as well as the methodological procedures.
  3. It is also useful to provide a thoughtful discussion of how the paper serves as a foundation for future research with a dedicated section on Directions for future Research.

On the basis of detailed comments sent to the authors, I propose a minor revision of the article prior to its publication.

Author Response

Reviewer #1:

Comments: The article entitled Nutrition in abrupt sunlight reduction scenarios: envisioning feasible balanced diets on resilient foods submitted for evaluation was written correctly and shows that a considerable effort went into your work. [...] On the basis of detailed comments sent to the authors, I propose a minor revision of the article prior to its publication.

Response: Thank you very much.

Comments: Is recommended to clarify what is (and why) resilient foods in the context of the study.

Response: We have included an explicit definition in S1P4L3-5

“A portfolio of food solutions that are resilient to abrupt catastrophic disruptions in the food system would be a valuable resource for increasing the resilience of food systems and preparedness globally. We define resilient foods or resilient food solutions as those foods, food production methods or interventions that would allow for significant food availability in the face of a global catastrophic food shock. These solutions should be well-suited for contributing to an adequate food supply for the greatest number of people even in the worst scenarios. Thus, this study focuses on technically-viable solutions that are both scalable and open source whenever possible to facilitate rapid production ramping. A preliminary version of such a resilient food portfolio is proposed in Section 2.3, with a focus on abrupt sunlight reduction scenarios.“

Comments: The model ASRS used in the study should be better explained in the "methods" as well as the methodological procedures.

Response: The model ASRS used in this study from Coupe et al. serves only as a guidance to better understand how the climate will change and where. Ongoing research is being done to better quantify the impact of sunlight reduction catastrophes on food systems, whose preliminary results inform the resilient food selection in the current work. We modified section 2.2 to better reflect this (S2.2P):

“Post-disaster scenario analysis was based on Coupe et al. [20,35] in-depth nuclear winter climate model. In their study, 150 Tg of soot are projected above the clouds, where rain cannot wash it out [20,36]. The subsequent climatic consequences such as large drops in temperature and precipitation levels were used as a basis to understand how the food system might be altered in this scenario, rather than as direct mathematical inputs. For example, it was used to approximate which foods could be more important such as crops capable of sustaining reasonably high yields with low temperatures and high water stress, or food sources independent of these conditions. The catastrophe strikes in May, which is likely a worst-case scenario. Models suggest that the climactic effect of particle emissions is larger during this period; additionally, food stocks are at their near lowest in the global North, as the Northern hemisphere harvest season has not started yet, and could potentially fail [36]. The results found using this model are also, to some extent, applicable to the volcanic winter scenario [20,35,37], even though the soot aerosols from nuclear fires result in more extreme climate effects.”

Comments: It is also useful to provide a thoughtful discussion of how the paper serves as a foundation for future research with a dedicated section on Directions for future Research.

Response: We have added additional explanation of the expected use of the presented analysis, including an example of its intended integration with upcoming research (S3.5P5):

"By combining the methodology presented in this study with analyses of specific catastrophic food scenarios based on concrete sets of assumptions, the specific forms of malnutrition that can be expected will be highlighted, as well as specific foods and interventions that could be implemented. Upcoming research will integrate this type of nutritional analysis with a parallel work analyzing food production, availability and affordability in specific ASRS scenarios."

Reviewer 2 Report

Dear Editor, dear authors,  

Overall, this is a well-written paper that relies on solid research in a relevant and timely topic: future-proofing our diets in the face of agrifood supply chain disruptions.

Methods used are adequate, and results are presented clearly.

As a scientist and engineer in this domain, at the intersection of risks and resilient food systems, I believe this paper stands to contribute to an expanding body of literature covering GCRs, future foods and alternative foods, future foods production systems, resilient food systems, and resilient diets.  

The paper is worthy of publication – however, not in its current form.

A few critical limitations and omissions should be resolved before a final decision could be made.     

First, the opening paragraph (lines 1-13 of the Introduction) – which is essential for explaining the motivation for this work, driving the argument, and providing the readers with much necessary context – draws too heavily (using almost the exact same words) on a different article published 7 months ago in the journal Nature Food, that is oddly neither mentioned nor cited here (Future foods for risk-resilient diets; https://www.nature.com/articles/s43016-021-00269-x). To avoid potential plagiarism, due recognition and reference must be made. 

In a similar vein, I would suggest to soften the statement: "currently,…little research on how to prepare for the most extreme and abrupt food system shocks". 

Furthermore, I suggest to include a brief definition of GCRs. 

In the paragraph: "This study aims at assessing the nutrition landscape in such a scenario, in order to better understand where planning and preparedness efforts should be focused to avoid massive famines, malnutrition and global public health issues." – I suggest to add "forms of malnutrition". 

In the paragraph: "A portfolio of food solutions…" – there is no such thing as "food solution", either say "food items" or "nutritional solutions".

In the paragraph, "Another key consideration is for these… alternative resilient foods…", as well as in section 2.3, section 2.3.2., section 2.4. and its subsections discussing "alternative resilient foods"  – authors should explain their choice of Seaweed, Single-cell protein from bacteria, and Lignocellulosic biomass. 

In lack of a better word, this limited set of items seems to me like 'cherry picking'.  

Why do authors ignore currently-available, indoors rearing of insects, a potential source of essential macro- and micro-nutrients; vitamins, minerals, essential fatty acids?  

Likewise, why do authors ignore currently-available, indoors cultivation of micro-algae in photo-bioreactors (PBRs), such as chlorella and nannochloropsis, a potential source of essential macro- and micro-nutrients; vitamins, minerals, essential fatty acids?

These unexplainable omissions reflect on the remaining of the article, and have implications for the analysis conducted in section 2.4.

I suggest to defend authors' choice in the article, or expand the analysis to include these items. 

Similarly, authors should correct the mistake of identifying seaweed as algae. It is macro-algae. 

Author Response

Reviewer #2:

Comments: Overall, this is a well-written paper that relies on solid research in a relevant and timely topic: future-proofing our diets in the face of agrifood supply chain disruptions.

Methods used are adequate, and results are presented clearly.

As a scientist and engineer in this domain, at the intersection of risks and resilient food systems, I believe this paper stands to contribute to an expanding body of literature covering GCRs, future foods and alternative foods, future foods production systems, resilient food systems, and resilient diets.

The paper is worthy of publication – however, not in its current form. A few critical limitations and omissions should be resolved before a final decision could be made.

Response: Thank you very much.

Comments: First, the opening paragraph (lines 1-13 of the Introduction) – which is essential for explaining the motivation for this work, driving the argument, and providing the readers with much necessary context – draws too heavily (using almost the exact same words) on a different article published 7 months ago in the journal Nature Food, that is oddly neither mentioned nor cited here (Future foods for risk-resilient diets; https://www.nature.com/articles/s43016-021-00269-x). To avoid potential plagiarism, due recognition and reference must be made.

Response: We apologize for this glaring oversight and have included the missing reference.

Comments: I would suggest to soften the statement: "currently,…little research on how to prepare for the most extreme and abrupt food system shocks".

Response: We have changed to: “There currently is a clear need for more work on preparing for the most extreme and abrupt food system shocks. ”

Comments: Furthermore, I suggest to include a brief definition of GCRs.

Response: We have added a typical definition of GCR and context on how it relates to global food security in S1P2L2-6.

“This would be a dramatic scenario, but unfortunately there exist even more severe food-related global catastrophic risks (GCRs) than the aforementioned. A common definition of a GCR is “an event that could damage human well-being on a global scale, even endangering or destroying modern civilization” [13]. The current agriculture-based global food system is known to function in “self-undermining, self-debilitating dynamics, disrupting yields and supply chains”, thus sustaining a significant vulnerability to global food catastrophe [5]. We classified a subset of these most extreme food catastrophes under the label of abrupt sunlight reduction scenarios (ASRS)”

Comments: In the paragraph: "This study aims at assessing the nutrition landscape in such a scenario, in order to better understand where planning and preparedness efforts should be focused to avoid massive famines, malnutrition and global public health issues." – I suggest to add "forms of malnutrition".

Response: The suggestion has been included in S1P3L11-13:

“This study aims at assessing the nutrition landscape in such a scenario, in order to better understand where planning and preparedness efforts should be focused to avoid massive famines, forms of malnutrition and global public health issues.”

Comments: In the paragraph: "A portfolio of food solutions…" – there is no such thing as "food solution", either say "food items" or "nutritional solutions".

Response: We have included an explicit definition of resilient food solution in S1P4L3-5

“A portfolio of food solutions that are resilient to abrupt catastrophic disruptions in the food system would be a valuable resource for increasing the resilience of food systems and preparedness globally. We define resilient foods or resilient food solutions as those foods, food production methods or interventions that would allow for significant food availability in the face of a global catastrophic food shock. These solutions should be well-suited for contributing to an adequate food supply for the greatest number of people even in the worst scenarios. Thus, this study focuses on technically-viable solutions that are both scalable and open source whenever possible to facilitate rapid production ramping. A preliminary version of such a resilient food portfolio is proposed in Section 2.3, with a focus on abrupt sunlight reduction scenarios. “

Here “food solution” refers to a food production method or intervention that can be used to address a problem, in this case a global food shock. For example:

  • Seaweed, spirulina and other SCPs, etc would count as resilient foods
  • Ramping up Seaweed/SCP/etc, or relocating cool tolerant crops would count as resilient food solutions in the sense that they are resilient food production methods.
  • Redirection of inputs initially assigned to biofuel and animal food production are examples of resilient food solutions that are neither a resilient food item nor a resilient food production method.

Though the latter are beyond the scope of the current research, we prefer to include them in the present definition for consistency, given their potential importance.

More background regarding the way we use the term “solutions” can be found here and here.

Comments: In the paragraph, "Another key consideration is for these… alternative resilient foods…", as well as in section 2.3, section 2.3.2., section 2.4. and its subsections discussing "alternative resilient foods"  – authors should explain their choice of Seaweed, Single-cell protein from bacteria, and Lignocellulosic biomass. In lack of a better word, this limited set of items seems to me like 'cherry picking'. Why do authors ignore currently-available, indoors rearing of insects, a potential source of essential macro- and micro-nutrients; vitamins, minerals, essential fatty acids?

Response: The Appendices hold justification for non-inclusion of insects. As it might be a question that many readers could ask themselves, we decided to put this content as appendices accessible at the end of the main document instead of in a separate supplement. We modified S2.3P1 accordingly:

“Resilient foods were selected upon their expected availability in the event of an ASRS. They include traditional foods such back, but also innovative resilient foods that are scalable and could provide a significant portion of the world's dietary needs. Affordability and caloric availability were also taken into consideration to exclude some potential food sources, such as mushrooms, insects, canola meal, fresh fruits and vegetables, nuts, synthetic fat, leaf protein concentrate and others; these are discussed in Appendix 4.”

Indeed, there are many other foods that could also be useful but were not included for various reasons. Regarding insects, the main argument is directly analogous to meat and animal products: animals are food converters, so while they could be nutritionally useful they do not contribute to primary calorie generation unless produced from inputs not digestible by humans. Based on the current analysis, it appears nutrition can be adequate in the catastrophe scenario without the need to resort to food converters, even though they could play a role as micronutrient sources. As a consequence, the animals of interest are cellulose-eating animals. For instance, this concerns ruminants, which are included here. However, the inclusion of cellulose-eating insects as resilient food sources in their current state does not appear to be sufficiently justified, particularly due to low affordability. More research is needed in this regard. We have added additional context in Appendix D, P3:

“Similarly, insects are currently costly to produce [18], and analogously to meat production most would be a net calorie sink. Those capable of digesting cellulose—whose feed does not compete with food for human consumption—appear to have limited potential to scale to significant global production in the proposed scenario for various reasons [16], as well as high cost [18]. However, they are also risk-resilient thanks to their potential to be scaled up over time and automated [125], and low sensitivity to extreme weather events if bred in stackable multi-compartment units [4]. More research would be needed to clarify affordability at scale and ramp-up potential.”

Comments: Likewise, why do authors ignore currently-available, indoors cultivation of micro-algae in photo-bioreactors (PBRs), such as chlorella and nannochloropsis, a potential source of essential macro- and micro-nutrients; vitamins, minerals, essential fatty acids?

Response: A combination of methanotrophic bacteria and spirulina was used in the nutritional analysis to represent the overall potential contribution of microbial foods to the diets. As with many other food categories, we used a limited number of examples to represent the category for simplicity, but this choice does not necessarily reflect the relative importance of each of the particular foods.

We have added additional context in S2.3.2P2L7-9:

“Certain single-cell organisms—some species of bacteria, microalgae and fungi—can be leveraged for production of protein-rich foods. Single-cell protein from agriculture-independent sources such as methane- or hydrogen-oxidizing bacteria could complement or replace animal proteins [58]. Ramping up this technology has the potential to make a significant contribution to fulfilling the protein requirements of the global population [27,29]. Here it is estimated that it would be available around one year after the catastrophe. Nutritional data from the methane-oxidizing bacteria of Unibio A/S was used as a proxy [59]. Spirulina (Arthrospira platensis) was used as well as a representative example of the group of cyanobacteria sometimes considered to be microalgae, including Spirulina, Chlorella and others.”

Also in S3.1.1.2:

“In a pessimistic scenario for Period I, it was proposed that stocks of staple crops (wheat, barley, canola, rice, corn, soy) would be unavailable and that most people would feed on potatoes, fish, cattle and sugar (Table 3). The diet proposed is, in this case, deficient in several micronutrients (Figure 2). Essential fatty acid intake was severely low (alpha-linolenic acid (ALA) and linoleic acid (LA)), which could be palliated by bacteria and microalgae-based foods. Fiber intake was as well, but this could be countered by consuming wholegrains, skins and husks. Protein intakes appear dangerously high. Vitamins D, E and K are again problematic, as well as manganese, magnesium and, to a lesser extent, potassium.”

And Appendix D, P9:

“The full potential of fermentation technologies to leverage a variety of microorganisms such as bacteria, fungi or microalgae for production of nutrient-rich foods was not included in this analysis. However, bacterial single cell protein from methane-oxidizing bacteria and from spirulina have been included as a representative example [137]. Similar microbial protein can be produced from CO2 and hydrogen [27]. Other emerging microbial foods often referred to as microalgae other than spirulina such as chlorella and nannochloropsis grown in photobioreactors could also bring useful nutritional contributions, such as fats and essential fatty acids. However, the energy efficiency values (electricity to calories) of cultivating these species in closed environments are expected to be much lower than for the abovementioned hydrogen-based single cell protein (4% vs 18%) [138], which also contain fats and essential fatty acids [139]. Microbial electrosynthesis could be used for production of acetic acid, and potentially longer-chain, more nutritionally-rich fatty acids as the technology develops [28]. Similarly, food ingredients such as sugars and glycerol (glycerin) could be produced with non-biological processes from agriculture-independent sources such as CO2 [140] or hydrocarbons [131].”

Comments: These unexplainable omissions reflect on the remaining of the article, and have implications for the analysis conducted in section 2.4. I suggest to defend authors' choice in the article, or expand the analysis to include these items.

Response: We hope to have sufficiently justified in the above responses why the suggested food sources, while useful in the proposed catastrophic scenario, were not included in the nutritional analysis. We have now moved the section including several of these other resilient foods from the supplement to the appendix and mention them more prominently in the main text, so as to better reflect their importance.

Comments: Similarly, authors should correct the mistake of identifying seaweed as algae. It is macro-algae.

Response: This has been changed in S2.3.2

“Seaweed (macroalgae)”

Round 2

Reviewer 2 Report

Considering the revised manuscript, additions, appendixes, and response to referees, and in light of the revised manuscript's soundness and significance, I fully support the publication of this paper, at the discretion of the chief editor.